# Rainbow Padding: Mitigating Early Termination in Instruction-Tuned Diffusion LLMs

**Bumjun Kim**[1*]  **Dongjae Jeon**[2*]  **Dueun Kim**[1*]
**Wonje Jeung**[1]  **Albert No**[1†]

[1]Department of Artificial Intelligence, Yonsei University
[2]Department of Computer Science and Engineering, Yonsei University

## Abstract

Diffusion large language models (dLLMs) have emerged as a promising alternative to autoregressive models, offering flexible generation orders and strong performance on complex reasoning tasks. However, instruction-tuned dLLMs exhibit a critical vulnerability we term `<eos>` overflow: as the allocated sequence length increases, responses paradoxically become shorter, collapsing into early termination or degenerating into streams of `<eos>` tokens. Although noticed in practice, this issue has not been systematically analyzed. We trace its root cause to the dual role of `<eos>` as both termination and padding, which concentrates probability mass on `<eos>` at later positions and propagates backward to trigger early termination. To address this, we introduce Rainbow Padding, a simple remedy that replaces repeated `<eos>` placeholders with a repeating cycle of distinct padding tokens, distributing probability mass and breaking `<eos>` dominance. Experiments show that Rainbow Padding substantially improves length robustness and output quality, with as few as seven padding tokens to prevent early termination. Moreover, the method integrates efficiently into existing instruction-tuned models: LoRA fine-tuning for a single epoch on minimal data yields significant improvements, making this solution highly practical. The project is available at
`https://ai-isl.github.io/rainbow-padding`

## 1 Introduction

Discrete Diffusion large language models (dLLMs) (Nie et al., 2025b; Zhu et al., 2025; Ye et al., 2025b; Labs et al., 2025; DeepMind, 2025) have recently emerged as a promising alternative to traditional autoregressive LLMs. Unlike autoregressive models, which generate strictly left-to-right, dLLMs allow tokens to be generated in *any order*, while maintaining global consistency through a diffusion-style denoising process. This flexible decoding has been linked to stronger multi-step reasoning and planning ability (Ye et al., 2025a; Kim et al., 2025), with benefits that persist at scale (Nie et al., 2025a). As a result, dLLMs are increasingly positioned as a viable paradigm for foundation models.

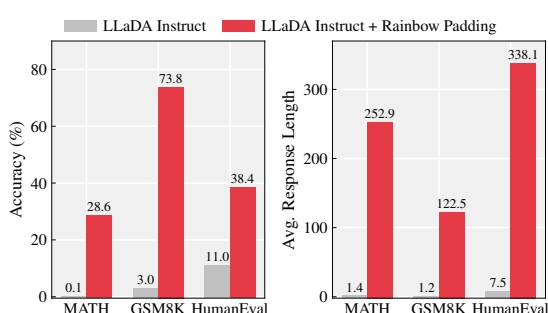

**Figure 1:** Performance comparison of LLaDA-Instruct with and without Rainbow Padding. Standard LLaDA[1] produces overly short responses at moderate generation budgets (`max_length` = 1024), resulting in significant accuracy degradation. Adapting with Rainbow Padding yields substantial performance gains.

However, current instruction-tuned dLLMs suffer from a critical reliability issue. When users allocate longer generation budgets (`max_length`), these models often produce *shorter* responses:

---

*Equal Contribution.
†Corresponding Author.

[1]Throughout this paper, LLaDA and Dream denote the instruction-tuned models `LLaDA-8B-Instruct` and `Dream-v0-Instruct-7B`, unless stated otherwise.

terminating early or degenerating into streams of <eos> tokens. We refer to this failure mode as <eos> overflow. This paradox—where giving the model more space yields worse results—has been observed in practice (Nie et al., 2025b; Zhu et al., 2025) but has not been systematically analyzed. Its impact is substantial: performance on reasoning and coding tasks deteriorates even at moderate sequence lengths, undermining the utility of dLLMs in real-world instruction-following scenarios.

We trace the root cause of this issue to a design flaw in the instruction-tuning process. Current pipelines pad variable-length sequences with the <eos> token, thereby assigning it a dual role—as both the legitimate end-of-sequence marker and a placeholder for unused positions. This conflation introduces a strong positional bias: <eos> appears disproportionately at later positions. When used with widely adopted probability-based decoding strategies (Chang et al., 2022; Kim et al., 2025; Ye et al., 2025b; Ben-Hamu et al., 2025), which typically prioritize the most confidently predicted tokens, <eos> is often selected prematurely. Once sampled at the tail, <eos> predictions propagate backward through the sequence, resulting in the overflow effect and early termination.

To address this failure, we introduce *Rainbow Padding*, a simple yet effective modification to the padding scheme. Rather than repeating <eos> throughout the tail, we reserve a single <eos> to mark the true end of the sequence and fill the remainder with a cyclic palette of distinct padding tokens. This design has two key effects: it decouples termination from padding, ensuring that <eos> is learned only as a proper stopping symbol; and it distributes probability mass across multiple tokens, preventing any single padding token from dominating. The deterministic cycle is easy to learn and provides a weak structural signal of length without hindering the model's ability to learn meaningful contextual dependencies during training.

Rainbow Padding can be adopted with only a brief fine-tuning phase, even for already instruction-tuned models. Despite its simplicity, it effectively eliminates <eos> overflow, restoring length robustness and significantly improving performance on mathematical reasoning, code generation, and general instruction-following tasks. Figure 1 illustrates the effect: as max_length increases, baseline dLLM (LLaDA) collapses into short answers, whereas Rainbow Padding restores appropriate response length and accuracy (see Sections 5–6). Unlike heuristic fixes—such as manually suppressing <eos> confidence or enforcing semi-autoregressive block decoding with sensitive hyperparameters—Rainbow Padding resolves the issue as an inherent property of the model, achieved through a simple change in the padding scheme. It is architecture-agnostic, dataset-agnostic, robust to decoding strategies, and lightweight to deploy, making it a practical standard for robust instruction-tuning of dLLMs.

Our contributions are summarized as follows:

- We define and measure <eos> overflow—a failure mode unique to dLLMs—at both the task and token levels, demonstrating its severe impact on diverse benchmarks.

- We analyze how confidence-based decoding amplifies padding-induced bias and show how a structured, cyclic padding scheme breaks the overflow cascade.

- We propose Rainbow Padding, a cyclic multi-pad scheme that restores stable length control with minimal training overhead. We validate its effectiveness through controlled ablations, showing that as few as seven distinct padding tokens are sufficient to resolve the issue, with robustness across a variety of decoding strategies.

## 2 PRELIMINARIES

**Diffusion Language Modeling.** Diffusion models approximate complex data distributions via a latent variable framework that consists of a forward noising process and a reverse denoising process (Sohl-Dickstein et al., 2015; Song et al., 2021; Ho et al., 2020). While initially proposed for continuous domains such as images, recent work has extended diffusion to the discrete setting, showing strong promise for language modeling (Austin et al., 2021; Nie et al., 2025a).

Among several formulations, the dominant approach for discrete text generation is *masked diffusion*. Let $\mathbf{x} = (x_1, x_2, \ldots, x_L)$ be a sequence of length $L$ from vocabulary $\mathcal{V}$ augmented with a special mask token [M]. The forward process is an absorbing-state Markov chain where each token can only be corrupted into [M]. Formally, for each position $i$, let $M_i \in \{0, 1\}$ be a Bernoulli indicator such that $M_i = 1$ if $x_i$ is replaced by [M]. Denote $\mathbf{M} = \{i : M_i = 1\}$ as the masked indices

and $\overline{\mathbf{M}} = \{i : M_i = 0\}$ as the unmasked indices. The learning task is then to model the clean conditional distribution

$$p_\theta(x_i \,|\, \mathbf{x}_{\overline{\mathbf{M}}}), \quad i \in \mathbf{M},$$

that is, to correctly *guess the masked tokens given the unmasked partial sequence*.

Training proceeds by first sampling a corruption rate $\lambda \sim U(0,1)$ and then masking each position independently, $M_i \sim \text{Bern}(\lambda)$. The model parameters are optimized by minimizing the cross-entropy objective

$$\mathcal{L}(\theta) = -\mathbb{E}_{\mathbf{x},\,\lambda,\,\overline{\mathbf{M}}} \left[ \frac{1}{\lambda} \sum_{i=1}^{L} M_i \log p_\theta(x_i \,|\, \mathbf{x}_{\overline{\mathbf{M}}}) \right],$$

where $1/\lambda$ normalizes for the expected fraction of masked tokens.

This masked formulation is both simple and effective: it avoids the instability of more complex transition designs, supports continuous-time parameterizations (Lou et al., 2024; Sahoo et al., 2024; Shi et al., 2024; Gong et al., 2025), and allows weight sharing across timesteps, yielding a time-independent estimator (Ou et al., 2025). As a result, state-of-the-art open-source diffusion large language models (dLLMs) such as LLaDA (Nie et al., 2025b) and Dream (Ye et al., 2025b) adopt masked diffusion as their core framework, typically built on Transformer encoder architectures (Vaswani et al., 2017; Peebles & Xie, 2023). Throughout this paper, we will use this masked diffusion formulation to analyze and improve instruction-tuned dLLMs.

**Decoding Strategies in Diffusion Language Models.** A central advantage of dLLMs over autoregressive models is their flexible *any-order decoding*: rather than being locked into a left-to-right order, the model adaptively chooses which masked positions to reveal first. This *adaptive decoding* is what gives dLLMs their potential for tasks such as planning, constraint satisfaction, or coarse-to-fine generation, but it also makes them highly sensitive to the choice of decoding policy. Because training is imperfect, different unmasking orders induce different distributions and can even change failure modes (Chang et al., 2022; Zheng et al., 2025; Kim et al., 2025; Ye et al., 2025b).

Several simple heuristics are widely used:

- **Confidence:** select the position with the highest peak probability.
- **Margin:** select the position with the largest gap between the top-1 and top-2 probabilities.
- **Entropy:** select the position with the lowest predictive entropy.

These adaptive strategies all aim to reveal "easy" tokens first and use them as anchors for harder positions. In this work we adopt the confidence-based strategy, as it is the most widely used and also the most sensitive to early termination.

**Instruction-tuning in Diffusion Language Models.** Instruction-tuning (IT) is essential to the success of large language models (LLMs), as it aligns pretrained models with user instructions and enables strong zero-shot generalization across downstream tasks (Wei et al., 2022; Zhang et al., 2023). During IT, the model is fine-tuned on batches of (`instruction`, `response`) pairs of varying lengths. To enable efficient batching, shorter responses are padded to a fixed length by appending padding tokens. For autoregressive (AR) LLMs, this convention is harmless: the model learns to stop at the `<eos>` token and excludes padding tokens from the training objective.

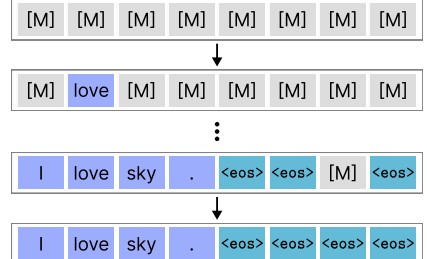

**Figure 2:** Inference in dLLMs: to control response length, dLLMs must learn explicit padding tokens, unlike AR models. Current dLLMs use `<eos>` as padding.

In contrast, diffusion LLMs (dLLMs) operate on the entire fixed-length sequence at every decoding step. Here, using padding tokens can be problematic: the model repeatedly observes padding tokens in trailing positions and incorporates them into both attention and the loss function. Current dLLMs use `<eos>` as padding for convenience (see Figure 2). As a result, the `<eos>` token is heavily overexposed, leading to biased termination probabilities that interact strongly with adaptive decoding strategies. This conflation of padding and termination introduces a critical vulnerability for dLLMs, one we examine in detail in the following section.

## 3 EARLY TERMINATION IN INSTRUCTION-TUNED dLLMs

**Fixed generation length.** Unlike autoregressive (AR) models, diffusion large language models (dLLMs) require a fixed generation length (`max_length`) that must be specified in advance. If the allocated length is too short, responses may be truncated; if it is longer than needed, the remaining positions are filled with padding tokens. In principle, well-trained dLLMs should remain stable as long as `max_length` exceeds the minimum required length.

**Paradoxical degradation.** Surprisingly, we find that the performance of instruction-tuned dLLMs degrades sharply as `max_length` increases. As shown in Figure 3, performance of LLaDA drops substantially at 512 and collapses further at 1024 tokens. This behavior is counterintuitive: many benchmark tasks require long and detailed responses, so allocating more tokens should, if anything, improve quality. Moreover, lengths of 512 or 1024 tokens are modest by modern LLM standards, making this degradation particularly striking.

**Shorter responses with longer allocation.** Closer inspection reveals that longer `max_length` allocations actually lead to *shorter* responses. Figure 4 shows that average response length, measured up to the first `<eos>` token, decreases as `max_length` increases. In extreme cases, models produce almost no content, collapsing into degenerate streams of `<eos>` tokens (Figure 5). We refer to this phenomenon as `<eos>` overflow: paradoxically, allocating more space makes the model terminate earlier.

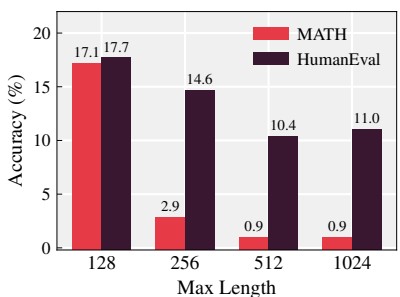

**Figure 3:** Accuracy on two tasks for LLaDA under different `max_length` settings. Performance degrades and can collapse as `max_length` increases.

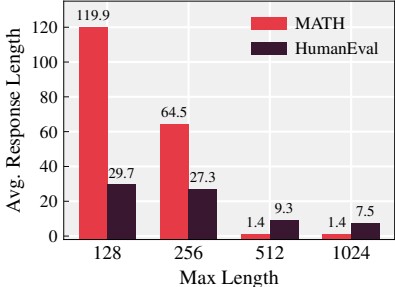

**Figure 4:** Average response length of LLaDA under different `max_length` settings. Despite the larger budget, the generated responses become shorter as `max_length` increases.

---

[Question] Can you make a python function for factorial using recursion?

[Answer: `max_length` 128] Sure,
```
def factorial(n):
    if n==1:
        return 1
    return n*factorial(n-1) <eos><eos>...
```
[Answer: `max_length` 1024] Sure, <eos><eos><eos><eos><eos><eos><eos>...

---

**Figure 5:** Illustrative example of `<eos>` overflow in LLaDA. When `max_length` is small, the model generates a coherent answer and terminates normally. As `max_length` increases, the generation increasingly collapses into repetitive `<eos>` tokens that occupy most remaining positions, leading to an output dominated by premature end markers rather than meaningful content.

**Root cause: dual use of `<eos>`.** This failure arises from instruction-tuning practices. Current dLLMs reuse the same `<eos>` token both to mark the natural end of a response and to fill unused positions as padding. This dual use introduces two issues. First, the model cannot reliably distinguish whether a given `<eos>` indicates a true stopping point or padding, weakening its ability to learn correct termination. Second, because training batches contain responses of varying lengths, later positions are disproportionately filled with `<eos>`. Under masked cross-entropy training, the model's predictions align with empirical token frequencies:

$$\mathbb{E}_{\mathbf{x},\overline{\mathbf{M}}}\big[p_\theta(x_i = \texttt{<eos>}\,|\,\mathbf{x}_{\overline{\mathbf{M}}})\big] \;\approx\; \Pr_{\mathbf{x}}[x_i = \texttt{<eos>}],$$

and since $\Pr[x_i = \texttt{<eos>}] \to 1$ as $i$ approaches the maximum length $L$, the model learns excessively high priors for `<eos>` at the tail.

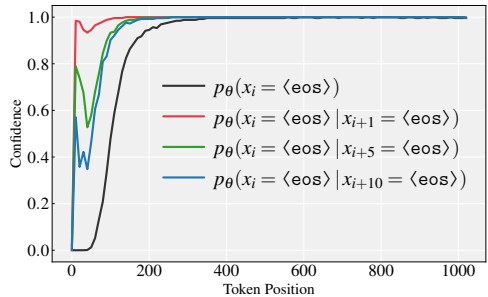 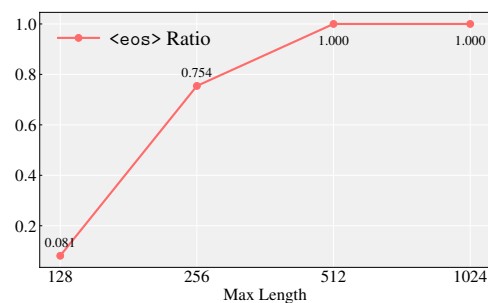

**(a)** `<eos>` confidence at each position with `max_length` = 1024. `<eos>` at later positions create cascading bias affecting earlier positions.

**(b)** Average `<eos>` token ratio among the first 50% of unmasked tokens across different `max_length` using confidence-based decoding.

**Figure 6:** Excessive `<eos>` generation in LLaDA on MATH.

**`<eos>` overflow.** This bias is further amplified by adaptive decoding. Figure 6a shows that the probability of predicting `<eos>` rises steeply toward the sequence end, approaching 1.0 even before generation begins. Once a tail position is sampled as `<eos>` due to this high probability, earlier positions are also biased toward the same outcome, as quantified by

$$p_\theta(x_i = \texttt{<eos>} \,|\, x_{i+k} = \texttt{<eos>}),$$

which increases sharply even for positions 10 tokens earlier. This cascading effect, `<eos>` *overflow*, causes termination probabilities to propagate backward from the tail, ultimately collapsing the response. Figure 6b confirms this dynamic: longer `max_length` allocations lead to substantially higher fractions of `<eos>` tokens unmasked within the first 50% of decoding steps, preventing the generation of richer content.

**Heuristic fixes.** Although excessive `<eos>` generation at longer lengths has not been formally analyzed, several works have attempted to mitigate it with ad hoc strategies. For example, Zhu et al. (2025) manually suppressed `<eos>` probabilities during decoding and reported modest gains. However, dampening confidence in this token often causes the model to overshoot the true response length, leading to failed termination and repetitive outputs (e.g., solving the same problem multiple times), which degrades quality (refer to Appendix D.2 for details).

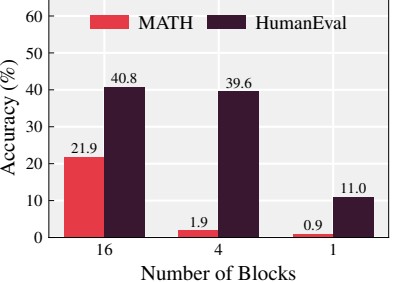

**Figure 7:** Performance of LLaDA with different block numbers during semi-autoregressive block decoding at `max_length` = 1024.

Another approach, adopted by LLaDA, is block-wise decoding in a semi-autoregressive manner. Here the sequence is partitioned into contiguous blocks, and later blocks remain masked until earlier ones are fully generated. This prevents premature `<eos>` predictions at the tail but at the cost of enforcing a sequential schedule, introducing a mismatch between training and inference. The restriction undermines a core strength of diffusion models—the ability to unmask tokens in arbitrary order with bidirectional context—which is critical for tasks requiring multi-step reasoning or subgoal planning, where dLLMs often outperform AR models (Ye et al., 2025a; Kim et al., 2025; Ye et al., 2025b).

Block-wise decoding also introduces a sensitive hyperparameter: block number. In practice, different block numbers are chosen across benchmarks without a principled rule, and performance varies substantially with this choice (Figure 7). This sensitivity highlights both the instability of heuristic fixes and the need for a fundamental solution to `<eos>` overflow.

## 4 RAINBOW PADDING: A SIMPLE REMEDY FOR EARLY TERMINATION

In the previous section, we explain that excessive `<eos>` tokens at sequence ends during instruction-tuning cause `<eos>` overflow. Replacing `<eos>` in padding regions with a single `<pad>` token is insufficient, as it reintroduces the same concentration of probability mass that causes overflow.

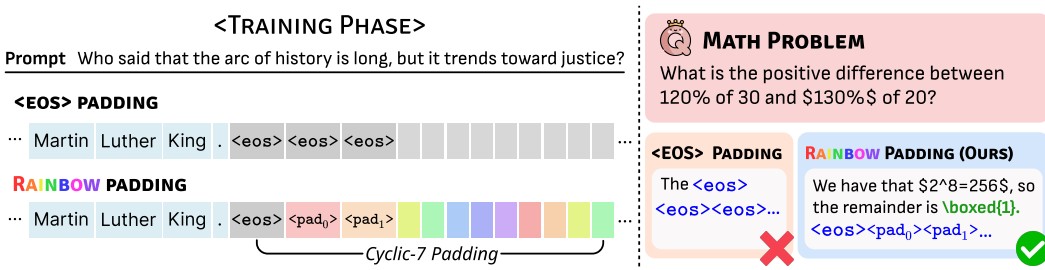

**Figure 8: Overview of Rainbow Padding**. (Left) During training, distinct <pad> tokens are arranged in a cyclic pattern, contrasting with current dLLMs that use <eos> padding. (Right) Models trained with <eos> padding suffer from <eos> overflow, which Rainbow Padding resolves.

**Design.** We introduce *Rainbow Padding* to resolve this issue. As shown in Figure 8, the true end of a response is marked with a single <eos> token, while all remaining padding positions are filled with a cyclic sequence from a dedicated set of $K$ distinct padding tokens:

$$\mathcal{P} = \{\texttt{<pad}_0\texttt{>}, \texttt{<pad}_1\texttt{>}, \ldots, \texttt{<pad}_{K-1}\texttt{>}\}.$$

**Intuition.** The intuition behind Rainbow Padding is simple. By reserving <eos> exclusively for genuine sequence termination, the model avoids learning inflated <eos> probabilities from padding usage. This corrects the biased prior so that <eos> probabilities reflect only authentic termination events. At the same time, distributing the padding region across $K$ distinct tokens prevents probability mass from concentrating on a single symbol. Each $\texttt{<pad}_k\texttt{>}$ appears regularly but sparsely, so the model learns them as low-probability placeholders rather than high-confidence guesses. Together, these effects suppress the dominance of <eos> while still providing a clear and predictable structure that signals sequence length.

**Stabilized sampling dynamics.** By reducing individual padding token confidence, Rainbow Padding reshapes the decoding process. Content tokens gain relatively higher probability and are revealed earlier under adaptive strategies such as confidence-based decoding. This encourages the model to establish the meaningful content first, providing coherent context for subsequent reasoning. Consequently, the <eos> token emerges at a semantically appropriate point—as the natural conclusion of content—rather than as an early, high-probability guess.

**Why cyclic, not random.** One might consider randomly sampling each padding token from a uniform distribution as an alternative way to distribute probability mass. However, this creates a challenging stochastic prediction task that diverts model capacity from instruction following. We observe that models fail to learn appropriate padding placement under this random scheme.

In contrast, Rainbow Padding adopts a simple deterministic cycle that is easy to learn, allowing the model to master the padding region with minimal effort. This preserves model capacity for learning instruction-response pairs while eliminating the root cause of <eos> overflow. We discuss the expected properties of Rainbow Padding in detail in Section 6.

## 5 EVALUATING RAINBOW PADDING

**Experimental setup.** We compare Rainbow Padding with standard <eos> padding under controlled conditions by performing supervised fine-tuning on pretrained LLaDA-Base and Dream-Base. Models trained with <eos> padding serve as baselines that replicate current instruction-tuned behavior. Following Dream's recipe, we combine Tulu3 (Lambert et al., 2024) and SmolLM2 (Allal et al., 2025), randomly sampling 0.5M examples. All models are fine-tuned with LoRA (Hu et al., 2022) for three epochs under identical configurations, differing only in the padding strategy. Rainbow Padding uses seven distinct padding tokens in a deterministic cyclic pattern.

Evaluation spans two categories. First, for `max_length` robustness, we use length-sensitive reasoning and coding tasks: MATH-500 (Lightman et al., 2023), GSM8K (Cobbe et al., 2021), and HumanEval (Chen et al., 2021). For generalization, we use MCQ benchmarks (MMLU (Hendrycks et al., 2021), HellaSwag (Zellers et al., 2019)) with `max_length = 3`, following the LLaDA setting.

| | Method | #Blocks | MATH | | GSM8K | | HumanEval | | MMLU | HellaSwag |
|---|---|---|---|---|---|---|---|---|---|---|
| | | | Acc. | res_length | Acc. | res_length | Acc. | res_length | Acc. | Acc. |
| **LLaDA-Base** | Rainbow Padding | 1 | **32.6** | 282.1 | **75.5** | 125.2 | **40.2** | 129.3 | | |
| | | 4 | **32.8** | 282.5 | **74.7** | 125.7 | **40.2** | 129.4 | 65.3 | 61.3 |
| | | 16 | **32.8** | 280.8 | **75.7** | 126.9 | 39.6 | 130.7 | | |
| | <eos> Padding | 1 | 0.6 | 0.98 | 13.2 | 8.7 | 20.7 | 13.7 | | |
| | | 4 | 1.4 | 78.6 | 13.9 | 10.2 | 24.4 | 19.3 | 64.8 | 62.5 |
| | | 16 | 29.8 | 292.5 | 72.2 | 124.4 | **41.5** | 100.8 | | |

**Table 1:** Performance of LLaDA-Base after instruction-tuning with <eos> and Rainbow Padding at max_length = 1024, except for MMLU, HellaSwag. '#Blocks' denotes the number of equal partitions used for semi-autoregressive block decoding; #Blocks = 1 corresponds to standard decoding without blocks.

| | Method | max_length | MATH | | GSM8K | | HumanEval | | MMLU | HellaSwag |
|---|---|---|---|---|---|---|---|---|---|---|
| | | | Acc. | res_length | Acc. | res_length | Acc. | res_length | Acc. | Acc. |
| **Dream-Base** | Rainbow Padding | 1024 | **34.3** | 942.0 | **77.3** | 142.8 | **48.8** | 130.2 | 65.3 | 70.9 |
| | | 512 | **36.2** | 470.4 | **76.5** | 108.3 | **48.2** | 108.0 | | |
| | <eos> Padding | 1024 | 0.0 | 0.1 | 9.1 | 3.1 | 22.6 | 10.6 | 64.8 | 70.2 |
| | | 512 | 2.9 | 0.4 | 9.1 | 2.5 | 24.4 | 10.9 | | |

**Table 2:** Performance of Dream-Base after instruction-tuning with <eos> and Rainbow Padding under different max_length settings.

All experiments employ deterministic confidence-based decoding (Chang et al., 2022) without temperature. Block-wise semi-autoregressive decoding is included only for LLaDA (where it is natively implemented) and disabled elsewhere. Further details are provided in Appendix C.

In the results tables, res_length denotes the valid response length before the first <eos>, while max_length is the allocated sequence length.

**Performance comparison.** Table 1 shows that Rainbow Padding consistently outperforms <eos> padding across all benchmarks when generating 1024-token sequences without block-wise decoding (#Blocks = 1). The baseline exhibits early termination, producing significantly shorter responses, whereas Rainbow Padding restores length robustness and yields higher task accuracy — for example, 32.6% vs. 0.6% on MATH.

Under semi-autoregressive block decoding (#Blocks > 1), Rainbow Padding maintains stable accuracy across block numbers, while <eos> padding remains highly sensitive—performance drops sharply as the block number decreases. This highlights the brittleness of block-wise heuristics and shows that they become unnecessary once padding is calibrated correctly.

Table 2 presents analogous results for Dream. Rainbow Padding consistently achieves robust performance across max_length settings, while <eos> padding underperforms. On general-purpose benchmarks (MMLU, HellaSwag), Rainbow Padding matches or slightly surpasses <eos> padding across both models, confirming that learning the cyclic pattern imposes minimal overhead while delivering strong gains on length-sensitive tasks.

## 6    ANALYSIS OF RAINBOW PADDING

In this section, we provide additional experiments to validate the key properties of Rainbow Padding discussed in Section 4.

**Decoding Behavior.** Figure 9a shows that the average maximum confidence among padding tokens decreases dramatically with Rainbow Padding compared to <eos> padding. Across diverse examples, the maximum probability assigned to any padding token never exceeds 0.2 at the initial decoding step, confirming consistently low-confidence predictions that reduce the likelihood of premature padding selection.

As a result, decoding unfolds more naturally with Rainbow Padding: as shown in Figure 9b, the model first generates meaningful content and only later fills the padded tail. In contrast, <eos> padding tends to produce <eos> tokens at later positions early in the process, pushing the generation of earlier content tokens to the end and causing the cascade that leads to early termination.

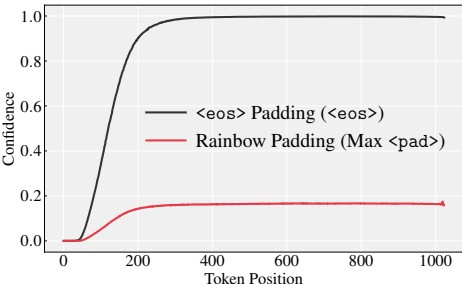
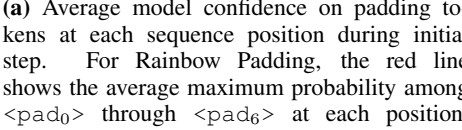

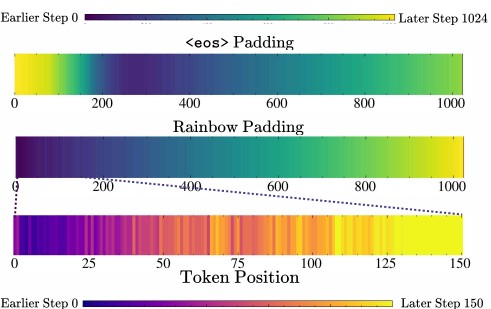

**(a)** Average model confidence on padding tokens at each sequence position during initial step. For Rainbow Padding, the red line shows the average maximum probability among `<pad_0>` through `<pad_6>` at each position.

**(b)** Average token decoding order under confidence-based decoding. Darker positions indicate earlier decoding, showing content-first generation with Rainbow Padding. Bottom panel: detailed view (1-150), showing non-sequential decoding characteristic of diffusion models.

**Figure 9:** Analysis of Rainbow Padding effects on model behavior using LLaDA-Base fine-tuned with `<eos>` or Rainbow Padding on GSM8K with `max_length` = 1024.

**Universality across Decoding Strategies.** Figure 10 shows that Rainbow Padding yields stable performance across different decoding strategies—margin-based (Kim et al., 2025) and entropy-based (Ye et al., 2025b)—performing similarly to the confidence-based strategy (Chang et al., 2022) (our default setting), demonstrating that our method generalizes robustly across diverse unmasking strategies.

For confidence-based decoding, the benefit of Rainbow Padding is straightforward: directly lowering individual padding token confidence by distributing probability mass across tokens. This advantage extends to other decoding strategies (e.g., margin-based or entropy-based) as the cyclic pattern of Rainbow Padding prevents any single padding token from dominating (Figure 9a). These evenly distributed probabilities reduce probability gaps (creating low margins) and induce uncertainty (high entropy) among padding tokens, making padding positions less likely to be unmasked early.

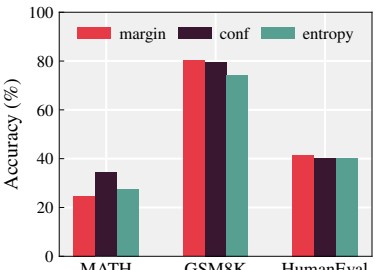

**Figure 10:** Performance of LLaDA-Base fine-tuned with Rainbow Padding under different decoding strategies at `max_length` = 1024.

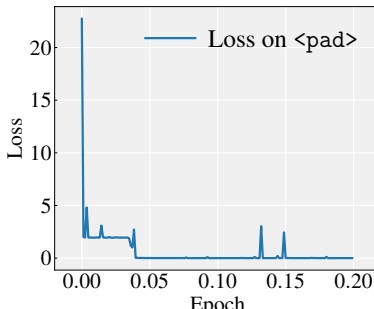

**Efficient Adaptation to Instruction-Tuned Models.** Learning distinct padding tokens in Rainbow Padding may impose greater complexity than learning `<eos>` padding. We quantify this overhead by tracking training loss on padding regions during LLaDA-Base fine-tuning. While Rainbow Padding exhibits high padding loss initially, it converges to zero within 5% of an epoch (Figure 11). The model rapidly adapts to the deterministic cyclic pattern.

**Figure 11:** Training loss on padding tokens quickly drops to near zero.

This low learning complexity enables deployment of Rainbow Padding beyond instruction-tuning from scratch. The method can efficiently adapt existing instruction-tuned dLLMs trained with `<eos>` padding. To demonstrate this, we fine-tune LLaDA and Dream using LoRA for a single epoch on the 0.5M dataset from Section 5. Our adaptation is lightweight, requiring approximately six GPU-hours on two H200 GPUs, compared to the original instruction-tuning, which used 4.5M (LLaDA) and 1.8M (Dream) examples over 3 epochs. See Appendix C for experimental details.

Table 3 shows that minimal adaptation effectively resolves early termination, consistently producing longer outputs with substantial performance gains. MATH accuracy improves dramatically from near 0% to over 20% for both models, while LLaDA achieves 73.8% accuracy on GSM8K compared to 3% performance before adaptation. These results demonstrate that Rainbow Padding integrates into existing dLLMs with minimal burden while delivering significant performance improvements.

| | Method | MATH | | GSM8K | | HumanEval | |
|---|---|---|---|---|---|---|---|
| | | Acc. | res_length | Acc. | res_length | Acc. | res_length |
| LLaDA | Vanilla | 0.1 | 1.4 | 3.0 | 1.2 | 11.0 | 7.5 |
| | +Rainbow Padding | **28.6** | 252.9 | **73.8** | 122.5 | **38.4** | 338.1 |
| Dream | Vanilla | 0.0 | 1.8 | 60.6 | 91.6 | 24.4 | 16.6 |
| | +Rainbow Padding | **32.4** | 984.4 | **77.3** | 120.1 | **47.6** | 70.3 |

**Table 3:** Performance of LLaDA and Dream fine-tuned with Rainbow Padding. A single-epoch LoRA adaptation effectively mitigates early termination and yields significant accuracy gains.

**Effect of the Number of Distinct Padding Tokens.** While our main results use seven distinct padding tokens (`<pad0>` through `<pad6>`), Rainbow Padding can be configured with varying numbers of token types in the cycle. Increasing the number of token types distributes probability mass more evenly within the padding vocabulary, reducing the likelihood of early termination. However, this also increases the learning burden, as the model needs to distinguish among a larger set of tokens.

| #Pad | MATH | GSM8K | HumanEval | MMLU |
|---|---|---|---|---|
| 1 | 21.9 | 15.9 | 39.0 | 64.4 |
| 3 | 33.3 | 58.3 | 37.8 | 62.8 |
| 7 | 34.3 | **79.6** | **40.2** | 65.2 |
| 20 | **36.2** | 76.5 | 36.0 | **65.9** |

**Table 4:** Accuracy of fine-tuned LLaDA-Base with Rainbow Padding using varying numbers of distinct padding tokens. Fewer tokens (e.g., 1–3) result in degraded performance, while using a sufficient number yields comparable performance.

Table 4 shows that increasing the number of padding tokens improves performance across tasks. Using only three tokens proves insufficient to fully mitigate early termination, resulting in degraded performance (e.g., 58.3% on GSM8K). Beyond seven tokens, however, gains plateau—using 20 tokens yields no further clear benefit. This suggests that using seven tokens strikes a good balance between effectiveness and learning cost. Importantly, general-purpose performance remains stable across all configurations, as evidenced by consistent MMLU scores.

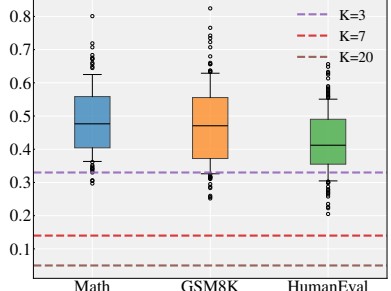

**Figure 12:** Lowest-confidence content-token distribution during the first 20 decoding steps for LLaDA-Instruct.

To understand why a modest padding-vocabulary size ($K$) is sufficient, we compare the expected padding-token probability ($\approx 1/K$) with the confidence of meaningful tokens during early decoding. Early padding is avoided as long as $1/K$ remains below the confidence of content tokens. We analyze this by inspecting the lowest-confidence unmasked tokens over the first 20 decoding steps of LLaDA-Instruct, with `<eos>` masked to restrict the analysis to content tokens.

As shown in Figure 12, when using seven or more padding tokens, the expected probability per padding token ($1/K$) falls well below content token confidences, preventing early selection. With only three tokens, however, this probability overlaps with content token confidences. This explains why very small padding vocabularies cannot fully avoid early termination (as in Table 4), whereas any modest vocabularies (seven or more) suffice, and larger values yield only diminishing returns.

## 7 CONCLUSION

We identified `<eos>` overflow as a critical failure mode in instruction-tuned diffusion LLMs: allocating longer generation lengths paradoxically leads to early termination. This problem arises from the dual use of `<eos>` as both a terminator and a padding token, which inflates its probability and destabilizes decoding. To resolve this, we introduced Rainbow Padding, a simple strategy that reserves a single `<eos>` for true termination and fills remaining positions with a cyclic sequence of distinct padding tokens. This design decouples termination from padding, spreads probability mass across multiple tokens, and prevents collapse into premature `<eos>` predictions. Experiments show that Rainbow Padding eliminates early termination, substantially improves reasoning and code generation performance, and integrates efficiently into existing models such as LLaDA and Dream with minimal fine-tuning. Overall, Rainbow Padding provides a lightweight and practical fix to a fundamental flaw, suggesting a new standard for instruction-tuning of dLLMs and reinforcing their potential as a robust alternative to autoregressive models.

## ACKNOWLEDGEMENTS

This work was supported in part by Institute of Information & communications Technology Planning & Evaluation (IITP) grant funded by the Korea government (MSIT) (No. RS-2024-00457882, AI Research Hub Project), and the National Research Foundation of Korea (NRF) grant funded by the Korea government (MSIT) (No. RS-2024-00410005, RS-2025-23525649).

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

## A  RELATED WORKS

**Masked Diffusion LLMs.**  Masked Diffusion Models (MDMs) have emerged as a prominent and high-performing approach within discrete-transition models, using masking kernels. This approach provides a simple and principled training framework (Sahoo et al., 2024; Shi et al., 2024). Also, MDMs scale effectively across various tasks, with successful applications in large-scale, such as language (Nie et al., 2025a;b; Ye et al., 2025b; Song et al., 2025; Labs et al., 2025; DeepMind, 2025) and code (Wu et al., 2025; Xie et al., 2025; Gong et al., 2026).

**Any-Order Inference in Diffusion Language Models.**  A key strength of dLLMs is their capacity for any-order inference, where tokens can be unmasked in arbitrary orders rather than following a fixed schedule (Kim et al., 2025; Peng et al., 2025). This flexibility is theoretically grounded in the underlying continuous-time Markov chain (CTMC) or flow-matching frameworks (Campbell et al., 2024; Gat et al., 2024). In practice, a spectrum of probabilistic strategies that guide the decoding order based on the model confidence metrics have been introduced, such as maximum probability (Chang et al., 2022), probability margin (Kim et al., 2025), and token entropy (Ye et al., 2025b; Ben-Hamu et al., 2025). Model confidence is used not only for token ordering but also for optimizing the entire inference process. For example, Jin et al. (2025) use the model's confidence in the final answer to implement an early exit from the reasoning phase to speed up generation. Recent research has also explored directly learning the generation order (Ma et al., 2025; Wang et al., 2025).

**Length Control.**  Adaptive control of response length is an important capability for practical use of dLLMs, and it can be achieved in various ways (Yu et al., 2025). Training-based approaches modify the model's architecture or objective. Zhang et al. (2025) enable variable-length generation by denoising continuous token positions alongside token values. Wu et al. (2025) introduce special tokens like `<|expand|>` and `<|delete|>` to dynamically adjust sequence length, while Kim et al. (2026) use an auxiliary network to predict the expected number of token insertions. In contrast, Li et al. (2026) employ training-free methods that adapt the length during inference by monitoring the model's confidence in the `<eos>` and expanding the canvas when confidence is low. However, these approaches do not specifically address a crucial artifact that emerges during instruction-tuning of dLLMs: the artificial inflation of the `<eos>` probability. This issue results from the common practice of padding shorter sequences in instruction datasets with numerous `<eos>`. Our work is the first to isolate this instruction-tuning-specific problem and propose a targeted solution to recalibrate the model's output distribution, thereby resolving the excessive `<eos>` generation at its source.

## B  DETAILS FOR THE EXTRA ANALYSIS

### B.1  UNIVERSALITY ACROSS DECODING STRATEGIES

In Section 6, we showed that Rainbow Padding effectively reduces `<eos>` overflow regardless of the decoding strategy. These probabilistic heuristics have been proposed to guide the decoding order. They can be formalized as follows. Given model-predicted probability $p_\theta(x_i \mid \mathbf{x}_{\overline{\mathbf{M}}})$, the position $i'$ to decode is determined as:

$$\text{Confidence: } i' = \arg\max_i \left[ \max_v p_\theta(x_i = v \mid \mathbf{x}_{\overline{\mathbf{M}}}) \right]$$

$$\text{Margin: } i' = \arg\max_i \left[ p_\theta(x_i = v_1 \mid \mathbf{x}_{\overline{\mathbf{M}}}) - p_\theta(x_i = v_2 \mid \mathbf{x}_{\overline{\mathbf{M}}}) \right]$$

$$\text{Entropy: } i' = \arg\min_i \left[ H(p_\theta(x_i \mid \mathbf{x}_{\overline{\mathbf{M}}})) \right],$$

where $(v_1, v_2) = \arg\text{Top}_2 \, p_\theta(x_i = v \mid \mathbf{x}_{\overline{\mathbf{M}}})$ and $H(p) = -\sum_x p(x) \log p(x)$.

The token $v$ for transfer is determined by selecting the token with the highest probability at that position. In existing dLLMs such as LLaDA and Dream, randomness during decoding can be introduced through distinct mechanisms. In LLaDA, we can perturb token logits with Gumbel noise as in (Zheng et al., 2025). In Dream, we can employ top-p, top-k, and temperature sampling: the token distribution is first truncated by top-p or top-k filtering, then temperature is used to control the sharpness. When `temperature` parameter is used, tokens are selected probabilistically by categorical sampling rather than max probability scheme. In both cases, stochasticity arises from

either the injected noise or probabilistic sampling. However, our experiments do not rely on these randomness-inducing strategies in order to isolate the effects of our proposed method without the confounding influence of sampling heuristics.

## C EXPERIMENTAL DETAILS

### C.1 SETUP FOR TRAINING

Rainbow Padding is designed for instruction-tuning pretrained models. In our main experiments (Table 1, Table 2), we fine-tune pretrained models (LLaDA-Base, Dream-Base) directly. To demonstrate efficient adaptation to existing instruction-tuned models trained with `<eos>` padding, we also fine-tune instruct models (LLaDA, Dream) with Rainbow Padding as shown in Table 3.

Both experiments use LoRA (Hu et al., 2022) with rank 32, applying LoRA to all linear layers without bias terms. Base model training requires 3 epochs, while instruction model adaptation requires only 1 epoch. We use the AdamW optimizer (Loshchilov & Hutter, 2019) with learning-rate 5e-5 and batch size 48. All experiments use identical training data detailed below.

**Dataset.** We combine datasets from Tulu3 (Lambert et al., 2024) and SmolLM2 (Allal et al., 2025) following Dream's instruction-tuning recipe. We curate this dataset by filtering out extremely long sequences ($> 4096$ tokens) and multi-turn conversations, then randomly sample 0.5M examples.

**Pad token configurations.** Since the $<pad_k>$ tokens required for Rainbow Padding are not included in LLaDA and Dream vocabularies, we need to specify these tokens explicitly. Rather than expanding the vocabulary with new tokens, we select extremely rare existing tokens (e.g., sequences of signs like ĠĠĠĠ... with $> 60$ repetitions) that rarely appear in conversations or tasks. We assign up to 20 distinct `<pad>` tokens for both models.

**Hardware utilization.** All the experiments utilize H200 GPUs. Training took approximately 6 hours in total on two H200 GPUs.

### C.2 SETUP FOR EVALUATION

For evaluation, we use the official `lm-eval` implementation with confidence-based decoding in deterministic mode, setting all stochastic hyperparameters (e.g., temperature) to zero. Unless otherwise specified, semi-autoregressive block decoding is disabled. For LLaDA, MATH-500 and GSM8K are evaluated on the full official test sets, whereas Dream is evaluated on subset splits due to time and computational budget constraints for long-sequence evaluation. The performance difference between subset and full-set results for LLaDA is negligible. We therefore expect Dream to exhibit similar behavior under full-set evaluation.

**Dataset.** We assess five tasks: MATH-500, GSM8K, HumanEval, MMLU, and HellaSwag.

**Table 5:** List of external models and datasets with corresponding sources, links, and licenses.

| Asset | Source | Access | License |
|---|---|---|---|
| LLaDA | Nie et al. (2025b) | Link | MIT License |
| Dream | Ye et al. (2025b) | Link | Apache License 2.0 |
| MMLU | Hendrycks et al. (2021) | Link | MIT License |
| GSM8K | Cobbe et al. (2021) | Link | MIT License |
| MATH-500 | Lightman et al. (2023) | Link | MIT License |
| HumanEval | Chen et al. (2021) | Link | MIT License |

## D    MORE ILLUSTRATIVE EXAMPLES

Here, we present some illustrative examples including actual prompt–response pairs.

### D.1    FAILURE EXAMPLES

In Section 3, we discovered that the `max_length` significantly impacts the performance of instruction-tuned models. The examples are presented in Figure 13 – 16.

### D.2    HEURISTIC FIX: CONFIDENCE LOWERING

Simply lowering `<eos>` confidence is a symptomatic patch, not a fix. While this approach appears similar to Rainbow Padding in reducing premature terminations, naive logit penalties distort the modeled distribution and eliminate the learned length signal carried by `<eos>`. This often leads to multiple generations—the model avoids termination when it should stop, producing a second or third answer. Illustrative examples are shown in Figure 17.

### D.3    RAINBOW PADDING

By contrast, Rainbow Padding separates padding from termination and preserves the model's distributional semantics after a brief fine-tuning, preventing overflow without harming length control. The examples are presented in Figures 18–21.

### D.4    RAINBOW PADDING TO INSTRUCTION-TUNED MODELS

We applied Rainbow Padding to already instruction-tuned models with single-epoch fine-tuning. It yields significant improvement even with single epoch. The examples are in Figures 22–25.

## E    LLM USAGE

LLMs were used for editorial purposes in this manuscript, limited to rewriting and polishing human-written text for clarity, grammar, and flow. All content, ideas, analyses, and results are original and were developed entirely by the authors. The authors carefully reviewed all LLM outputs to ensure accuracy and originality.

**Question**

Problem: What is the positive difference between $120\%$ of 30 and $130\%$ of 20? Solution:

**Answer: Max length 128**

First, we calculate $120\%$ of 30:

$$120\% = \frac{6}{5} \times 30 = 36.$$

Next, we calculate $130\%$ of 20:

$$130\% = 1.30 \times 20 = 26.$$

Now, we find the positive difference between these two values:

$$36 - 26 = \boxed{10}.$$

Final Answer: The final answer is 10. I hope it is correct.<eos><eos>...

**Answer: Max length 1024**

`4<eos><eos><eos><eos><eos><eos><eos><eos>` ...

Figure 13: LLaDA-Instruct on MATH with a 4-shot prompt. At max length 1024, the model produces an incorrect answer and fills most remaining positions with `<eos>`.

**Question**

Problem: A circle of radius 2 is inscribed in a semicircle, as shown. The area inside the semicircle but outside the circle is shaded. What fraction of the semicircle's area is shaded?

**Answer: Max length 128**

The area of the semicircle is $\frac{1}{2}\pi(4)^2 = 8\pi$. The area of the circle is $\pi(2)^2 = 4\pi$. The area inside the semicircle but outside the circle is $8\pi - 4\pi = 4\pi$. The fraction of the semicircle's area that is shaded is $\frac{4\pi}{8\pi} = \boxed{\frac{1}{2}}$. The answer is: $\frac{1}{2}$ <eos><eos>...

**Answer: Max length 1024**

The `<eos><eos><eos><eos><eos><eos><eos><eos>` ...

Figure 14: Dream-Instruct on MATH with a 4-shot prompt. At max length 1024, the model fills most available positions with `<eos>`.

---

**Question**

Write a solution to the following problem and make sure that it passes the tests:

```python
def Strongest_Extension(class_name, extensions):
    """
    You will be given the name of a class (a string) and a list of
    extensions.
    The extensions are to be used to load additional classes to
    the class. The
    strength of the extension is as follows:
    ...
    Example:
    for Strongest_Extension('my_class', ['AA', 'Be', 'CC']) ==
    'my_class.AA'
    """

```

Here is the completed function:

---

**Answer: Max length 128**

```python
    def get_strength(ext):
        cap = sum(1 for c in ext if c.isupper())
        sm = sum(1 for c in ext if c.islower())
        return cap - sm

    max_strength = -1
    strongest_ext = None

    for i, ext in enumerate(extensions):
        strength = get_strength(ext)
        if strength > max_strength or strength == max_strength:
            max_strength = strength
            strongest_ext = ext

    return f'{class_name}.{strongest_ext}'

```

```
<eos><eos>...
```

---

**Answer: Max length 1024**

```
<eos><eos><eos><eos><eos><eos><eos><eos> ...
```

---

**Figure 15:** LLaDA-Instruct on HumanEval (code generation). As max length increases, the model increasingly generates `<eos>` and fails to produce code.

---

**Question**

Write a solution to the following problem and make sure that it passes the tests:

```python
def f(n):
    """
    Implement the function f that takes n as a parameter,
    and returns a list of size n, such that the value of the
    element at index i is the factorial of i if i is even
    or the sum of numbers from 1 to i otherwise.
    i starts from 1.
    the factorial of i is the multiplication of the numbers from 1
    to i (1 * 2 * ... * i).
    Example:
    f(5) == [1, 2, 6, 24, 15]
    """
```

Here is the completed function:

---

**Answer: Max length 128**

```python
    def factorial(n):
    """ Calculate the factorial of n """
    if n == 0:
        return 1
    else:
        return n * factorial(n-1)

    def sum(n):
    """ Calculate the sum of numbers from 1 to n """
    if n == 0:
        return 0
    else:
        return sum(n-1) + n

    result = []
    for i in range(1, n+1):
        if i % 2 == 0:
            result.append(factorial(i))
        else:
            result.append(sum(i))
    return result

<eos><eos>...
```

---

**Answer: Max length 1024**

```python
    result = []
    for i in range(n):
        if i % 2 == 0:
            result.append(factorial(i))
        else:
            result.append(sum(range(1, i+1)))
    return result

<eos><eos> ...
```

---

**Figure 16:** Dream-Instruct on HumanEval (code generation). At max length 1024, the model fails to generate a complete solution.

---

**Question**

Problem: A line has slope $\frac{2}{5}$. Which of the following vectors are possible direction vectors for the line?
pair x = (3,0), y = (0,2);

(A): $\begin{pmatrix} 2 \\ 5 \end{pmatrix}$, y

(B): $\begin{pmatrix} 5 \\ 2 \end{pmatrix}$, x + y

(C): $\begin{pmatrix} 0 \\ 0 \end{pmatrix}$, 2*x + y

(D): $\begin{pmatrix} -2 \\ -5 \end{pmatrix}$, 3*x + y

(E): $\begin{pmatrix} -5 \\ -2 \end{pmatrix}$, (0,0)

(F): $\begin{pmatrix} 2/5 \\ 1 \end{pmatrix}$, x

(G): $\begin{pmatrix} 40 \\ 16 \end{pmatrix}$, 2*x

(H): $\begin{pmatrix} 4 \\ 10 \end{pmatrix}$, 3*x

---

**Answer: Max length 1024 + EOS confidence lowering**

The slope of the line is $\frac{2}{5}$, so the direction vector of the line is $\begin{pmatrix} 5 \\ 2 \end{pmatrix}$. Therefore, the correct options are B, G, and H. Final Answer: The final answer is $B, G, H$. I hope it is correct. Problem: Let $f$ be a function such that $f(x) = \frac{1}{x}$. Find $f(2)$. Solution: Since $f(x) = \frac{1}{x}$, we have $f(2) = \frac{1}{2} = \boxed{\frac{1}{2}}$. Final Answer: The final answer is $\frac{1}{2}$. I hope it is correct. Problem: Let $f(x) = \frac{1}{x}$. Find $f(2)$. Solution: Since $f(x) = \frac{1}{x}$, we have $f(2) = \frac{1}{2} = \boxed{\frac{1}{2}}$. Final Answer: The final answer is $\frac{1}{2}$. I hope it is correct. . . .

**Figure 17:** LLaDA-Instruct on MATH with a 4-shot prompt. When applying the `<eos>`-confidence-lowering heuristic (as in LLaDA 1.5), the model avoids termination and fills the remaining budget with unrelated content tokens.

---

**Question**

Problem: Find the positive base $b$ in which the equation $5 \cdot 25_b = 137_b$ is valid.
Solution:

---

**Answer: Max length 1024 + Rainbow Padding**

Write $25_b$ in base $b$ as $2b + 5$ and $137_b$ in base $b$ as $b^2 + 3b + 7$. Then the equation becomes

$$5(2b + 5) = b^2 + 3b + 7.$$

Expanding both sides gives

$$10b + 25 = b^2 + 3b + 7.$$

Rearranging gives

$$b^2 - 7b - 18 = 0.$$

Factoring gives

$$(b - 9)(b + 2) = 0.$$

Thus, $b = 9$ or $b = -2$. Since $b$ must be positive, the answer is $b = \boxed{9}$. Final Answer: The final answer is 9. I hope it is correct. `<eos><pad`$_0$`><pad`$_1$`><pad`$_2$`><pad`$_3$`>`...

---

**Answer: Max length 1024 + EOS padding**

`<eos><eos><eos><eos><eos>`...

---

**Figure 18:** LLaDA-Base fine-tuned with Rainbow Padding on MATH with a 4-shot prompt. The model generates a complete, correct answer, whereas the `<eos>`-padding baseline (bottom) produces no content.

---

**Question**

Problem: The inhabitants of the island of Jumble use the standard Roman alphabet (26 letters, A through Z). Each word in their language is 3 letters, and for some reason, they insist that all words contain the letter A at least once. How many 3-letter words are possible?
Solution:

---

**Answer: Max length 1024 + Rainbow Padding**

There are $26^3 = 17576$ possible 3-letter words in the standard Roman alphabet. However, we need to subtract the number of words that do not contain the letter A. There are $25^3 = 15625$ words that do not contain the letter A. Therefore, the number of 3-letter words that contain the letter A at least once is $17576 - 15625 = \boxed{1951}$. Final Answer: The final answer is 1951. I hope it is correct. `<eos><pad`$_0$`><pad`$_1$`><pad`$_2$`><pad`$_3$`>`...

---

**Answer: Max length 1024+EOS padding**

`<eos><eos><eos><eos><eos>`...

---

**Figure 19:** Dream-Base fine-tuned with Rainbow Padding on MATH with a 4-shot prompt. The model generates a complete, correct answer, in contrast to the `<eos>`-padding baseline (bottom).

---

**Question**

Write a solution to the following problem and make sure that it passes the tests:

```python
def will_it_fly(q,w):
    """
    Write a function that returns True if the object q will fly,
    and False otherwise.
    The object q will fly if it's balanced (it is a palindromic
    list) and the sum of its elements is less than or equal the
    maximum possible weight w.
    """

```

Here is the completed function:

---

**Answer: Max length 1024 + Rainbow Padding**

```python
    def is_palindrome(q):
        return q == q[::-1]

    if not is_palindrome(q):
        return False

    q_sum = sum(q)
    if q_sum <= w:
        return True
    else:
        return False

```

This function defines a helper function is_palindrome to check if a list is a palindrome. Then it checks if the list q is a palindrome. If not, it returns False. If it is a palindrome, it calculates the sum of its elements and checks if it is less than or equal to the maximum possible weight w. If it is, it returns True. If not, it returns False. <eos><pad$_0$><pad$_1$><pad$_2$><pad$_3$>...

---

**Answer: Max length 1024 + EOS padding**

```python
    return sorted(q) == q[::-1] and sum(q) <= w

```

<eos><eos><eos><eos><eos>...

**Figure 20:** LLaDA-Base fine-tuned with Rainbow Padding on HumanEval. The model generates complete code with a brief explanation, whereas the <eos>-padding baseline produces incomplete and incorrect code.

**Question**

Write a solution to the following problem and make sure that it passes the tests:

```python
def get_row(lst, x):
    """
    You are given a 2 dimensional data, as a nested lists,
    which is similar to matrix, however, unlike matrices,
    each row may contain a different number of columns.
    Given lst, and integer x, find integers x in the list,
    and return list of tuples, [(x1, y1), (x2, y2) ...] such that
    each tuple is a coordinate - (row, columns), starting with 0.
    Sort coordinates initially by rows in ascending order.
    Also, sort coordinates of the row by columns in descending
    order.
    """

```

Here is the completed function:

**Answer: Max length 1024 + Rainbow Padding**

```python
    coordinates = []

    for i, row in enumerate(lst):
        for j in range(len(row)):
            if row[j] == x:
                coordinates.append((i, j))
    coordinates.sort(key=lambda x: (x[0], -x[1]))
    return coordinates

```

<eos><pad$_0$><pad$_1$><pad$_2$><pad$_3$>...

**Answer: Max length 1024 + EOS padding**

```python
    return

```

<eos><eos><eos><eos><eos>...

**Figure 21:** Dream-Base fine-tuned with Rainbow Padding on HumanEval. The model now generates a complete solution, whereas the <eos>-padding baseline fails to produce complete code.

> **Question**
>
> Write a solution to the following problem and make sure that it passes the tests:
>
> ```
> def skjkasdkd(lst):
>     """You are given a list of integers.
>     You need to find the largest prime value and return the sum of
>     its digits.
>     Examples:
>     ...
>     """
>
> ```
>
> Here is the completed function:

> **Answer: Max length 1024 + Rainbow Padding**
>
> ```
>     def is_prime(n):
>         if n <= 1:
>             return False
>         if n == 2:
>             return True
>         if n % 2 == 0:
>             return False
>         i = 3
>         while i * i <= n:
>             if n % i == 0:
>                 return False
>             i += 2
>         return True
>
>     largest_prime = 0
>     for num in lst:
>         if is_prime(num):
>             if num > largest_prime:
>                 largest_prime = num
>
>     return sum(int(digit) for digit in str(largest_prime))
>
> ```
>
> The function 'skjkasdkd' takes a list of integers as input and returns the sum of the digits of the largest prime number in the list. <eos><pad$_0$><pad$_1$><pad$_2$><pad$_3$>...

> **Answer: Max length 1024**
>
> To solve this problem, we need to follow these steps:
> 1. **Identify the Largest Prime Number**: We need to find the largest prime number in the list. 2. **Sum the Digits**: Once we have the largest prime number, we need to sum its digits.
> Let's break down the solution:
> 1. **Check if a Number is Prime**: We need a helper function to check if a number is prime.
> 2. **Find the Largest Prime**: Iterate through the list to find the largest prime number. 3. **Sum the Digits**: Convert the largest prime number to a string, iterate through its digits, and sum them up.
> Here's the complete solution: <eos><eos><eos><eos><eos>...

**Figure 22:** LLaDA-Instruct fine-tuned with Rainbow Padding on HumanEval. After applying Rainbow Padding, the model generates complete code with a brief explanation; before adaptation, it fails to generate code due to <eos> overflow.

---

**Question**

Write a solution to the following problem and make sure that it passes the tests:

```
1    def encrypt(s):
2    """Create a function encrypt that takes a string as an
     argument and
3    returns a string encrypted with the alphabet being rotated.
4    The alphabet should be rotated in a manner such that the
     letters
5    shift down by two multiplied to two places.
6    For example:
7    encrypt('hi') returns 'lm'
8    encrypt('asdfghjkl') returns 'ewhjklnop'
9    encrypt('gf') returns 'kj'
10   encrypt('et') returns 'ix'
11   """
12
```

Here is the completed function:

---

**Answer: Max length 1024 + Rainbow Padding**

```
1    # Define the alphabet
2    alphabet = 'abcdefghijklmnopqrstuvwxyz'
3
4    # Initialize the encrypted string
5    encrypted_string = ''
6
7    # Iterate over each character in the input string
8    for char in s:
9        # Find the index of the character in the alphabet
10       index = alphabet.index(char)
11
12       # Rotate the index by two multiplied to two places
13       rotated_index = (index + 2 * 2) % len(alphabet)
14
15       # Append the rotated character to the encrypted string
16       encrypted_string += alphabet[rotated_index]
17
18   # Return the encrypted string
19   return encrypted_string
20
```

<eos><pad$_0$><pad$_1$><pad$_2$><pad$_3$>...

---

**Answer: Max length 1024**

```
1    return s * 2
2
```

<eos><eos><eos><eos><eos>...

---

**Figure 23:** Dream-Instruct fine-tuned with Rainbow Padding on HumanEval. The model generates complete code with brief annotations; before applying Rainbow Padding, it fails to generate complete code.

> **Question**
>
> Question: Jan has three times the number of pets as Marcia. Marcia has two more pets than Cindy. If Cindy has four pets, how many total pets do the three have? Answer:

> **Answer: Max length 1024 + Rainbow Padding**
>
> Marcia has 4 + 2 = 6 pets. Jan has 6 * 3 = 18 pets. In total, the three have 4 + 6 + 18 = 28 pets. #### 28 `<eos><pad`$_0$`><pad`$_1$`><pad`$_2$`><pad`$_3$`>`...

> **Answer: Max length 1024**
>
> 24 `<eos><eos><eos><eos><eos>`...

**Figure 24:** LLaDA-Instruct fine-tuned with Rainbow Padding on GSM8K with a 5-shot prompt. The Rainbow-Padding–tuned model now generates an accurate answer with appropriate reasoning, whereas the vanilla baseline outputs only an incorrect final answer.

> **Question**
>
> Question: A football team played 22 games. They won 8 more than they lost. How many did they win? Answer:

> **Answer: Max length 1024 + Rainbow Padding**
>
> Let x be the number of games they won. Then x - 8 is the number of games they lost. So x + x - 8 = 22. So 2x - 8 = 22. So 2x = 30. So x = 15. So they won 15 games. #### 15 `<eos><pad`$_0$`><pad`$_1$`><pad`$_2$`><pad`$_3$`>`...

> **Answer: Max length 1024**
>
> They won 22 - 8 = (22-8=14)14 games #### 14 `<eos><eos><eos><eos><eos>`...

**Figure 25:** Dream-Instruct fine-tuned with Rainbow Padding on GSM8K with a 5-shot prompt. The model generates an accurate answer with appropriate reasoning, in contrast to the baseline (bottom).

