# OpenReview forum: "Rainbow Padding: Mitigating Early Termination in Instruction-Tuned Diffusion LLMs"
_ICLR.cc/2026/Conference — ICLR 2026 Poster_

### Official Review · Reviewer_rf99 · 2025-10-27

**Soundness:** 3
**Presentation:** 3
**Contribution:** 3
**Rating:** 6
**Confidence:** 4

**Summary:**

This work addresses the <eos> overflow phenomenon in instruction-tuned diffusion-based large language models (dLLMs). The authors identify the root cause is that the <eos> is both used as padding token and sequence terminator, which lead to overestimated probability of <eos> at later positions. The authors propose Rainbow Padding to replace <eos> padding tokens with a cyclical sequence of distinct padding tokens. Experiments are conducted to prove that the Rainbow Padding gains overall improvements on various benchmarks with minimal training cost.

**Strengths:**

1. The paper defines and analyzes the <eos> overflow issue, which has practical implications for deploying dLLMs in instruction-following scenarios.
2. Rainbow Padding is intuitive, easy to implement, and does not require architectural changes or complex decoding strategies.
3. Comprehensive experiments are conducted on multiple benchmarks with insightful analysis.

**Weaknesses:**

1. The paper lacks formal analysis or modeling of why Rainbow Padding works, especially in terms of training dynamics or probabilistic behavior.
2. The use of 7 distinct padding tokens is empirically justified, but no principled method or adaptive mechanism is provided for selecting this number.

**Questions:**

1. Have you considered adaptive methods for selecting the number of padding tokens?
2. Will Rainbow Padding introduce additional bias in length control, such as over-generation or difficulty in terminating properly?
3. Have you validate your methods on some larger LLM architectures?

---

> ### Author Response · Authors · 2025-11-19
> **Response to Reviewer rf99 (1/2): W1, W2, Q1**
>
> **Response to W1 & W2 & Q1: Lacks formal analysis of why Rainbow Padding mechanism works, and provides no principles or adaptive method for choosing $K$.**
>
>
> 1. Rainbow Padding mechanism: its cyclic pattern distributes probability evenly across padding tokens
>
> Rainbow Padding prevents early termination by ensuring that no single formatting token (\<eos> or \<pad>) dominates the probability distribution during generation. By cycling through $K$ distinct padding tokens, the model avoids repeatedly encountering the same padding region, preventing any single token from becoming overly probable. This distributes the overall padding probability across the entire cycle, keeping each padding token’s probability close to $1/K$ (see Figure 9(a)).
>
> 2. $K$ is not a sensitive hyperparameter: any moderate large $K$ is reasonable
>
> Any sufficiently large $K$ prevents premature termination as long as $1/K$ remains below the confidence assigned to meaningful content tokens. As shown in Table 4, increasing $K$ beyond a modest point (e.g., $K=7$, $K=20$) results in negligible difference: once $1/K$  is small enough, padding tokens can no longer dominate the distribution. This explains the saturation effect and why the padding vocabulary size does not require fine-grained tuning.
>
> 3. Empirical evidence on $K$ selection
>
> To illustrate this more concretely, we measure the lowest confidence assigned to unmasked content tokens during the first 20 generation steps of LLaDA-Instruct, while making \<eos> to be unselectable. Across tasks, content-token confidence stays well above 0.20:
>
> | **Dataset** | **1st Percentile** | **10th Percentile** | **Mean** | **90th Percentile** |
> |-------------|--------------------|----------------------|----------|----------------------|
> | MATH    | 0.305              | 0.362                | 0.486    | 0.637                |
> | GSM8K   | 0.257              | 0.326                | 0.472    | 0.632                |
> | HumanEval  | 0.225            | 0.303                | 0.422    | 0.554                |
>
> For Rainbow Padding to prevent early termination, we require $1/K$ to be below this range.
>
> Thus:\
> $K=7  →  1/K \approx 0.14$ (safe),\
> $K=20  →  1/K \approx 0.05$ (safe),\
> $K=3 → 1/K \approx 0.33$ (unsafe, overlaps with content-token confidence).
>
> This analysis explains why $K=3$ frequently fails but $K≥7$ is consistently effective, and this saturation behavior observed on LLaDA is expected to generalize to other dLLMs as well. We have added this discussion and the new experimental results to the revised manuscript in (Figure 12, lines 476–485 on page 9).

---

> > ### Author Response · Authors · 2025-11-19
> > **Response to Reviewer rf99 (2/2): Q2, Q3**
> >
> > **Response to Q2: Does Rainbow Padding introduce additional bias in length control, such as over-generation or difficulty in proper termination?**
> >
> > Thank you for the valuable question. We empirically found that Rainbow Padding is less likely to induce the length bias that prevents proper response termination.
> > To illustrate this, we compare the average response lengths of LLaDA-base trained with Rainbow Padding against standard \<eos> padding with a 16-block decoding heuristic (LLaDA’s default strategy to avoid \<eos> overflow)
> >
> > | **Dataset**   | **Rainbow Padding ($K=7$)** | **\<eos> Padding (16-block decoding)** |
> > |-----------|-------------------------|-----------------------------------|
> > | MATH      | 285.7                  | 305.4                             |
> > | GSM8K     | 115.7                  | 119.4                             |
> > | HumanEval | 129.3                  | 100.8                             |
> >
> > These results show that Rainbow Padding achieves comparable response lengths to heuristic-based approaches without requiring any external fixes. The model inherently learns to produce well-calibrated response lengths through training.
> >
> > **Response to Q3: Have you validated Rainbow Padding on some larger LLM architectures?**
> >
> >
> > Thank you for this important question. We validated Rainbow Padding on LLaDA-8B and Dream-7B, two of the most representative large-scale open-source dLLMs currently available. While additional dLLMs exist, many are closed-source (e.g., Mercury), task-specific (e.g., DiffuCoder-7B), or direct variants of LLaDA (e.g., LLaDA 1.5, MMaDA), which we expect to exhibit similar \<eos>-overflow behavior. Since Rainbow Padding is an architecture-agnostic training strategy, it can be applied independently of model scale.
> >
> > In addition, we conducted supplementary experiments to assess stability of Rainbow Padding  under longer allocated generation lengths (i.e., max_length = 2048). Results for LLaDA are reported below.
> >
> > | **Dataset** | **Metric**         | **max_length = 1024 (our default)** | **max_length = 2048** |
> > |-------------|---------------------|-------------------------|-------------------------|
> > | MATH        | Acc.            | 34.3                     | 33.3                     |
> > | MATH        | Response Length     | 285.7                    | 385.9                     |
> > | GSM8K       | Acc.            | 79.6                     | 75.8                    |
> > | GSM8K       | Response Length     | 115.7                     | 118.1                    |
> >
> > We observe that Rainbow Padding remains stable even when the allocated length budget is doubled to 2048 tokens, indicating that the model effectively learned proper termination behavior. Given this mechanism and its empirical robustness, we expect Rainbow Padding to extend naturally to larger dLLM architectures.

---

### Official Review · Reviewer_f5z9 · 2025-10-30

**Soundness:** 3
**Presentation:** 4
**Contribution:** 3
**Rating:** 6
**Confidence:** 3

**Summary:**

The paper identifies a critical failure phenomenon in instruction-tuned diffusion LLMs (dLLMs), termed eos overflow. This phenomenon describes a performance degradation where a longer generation length leads to prematurely short or degenerated responses. The authors trace the root cause to the dual use of the eos token for both sequence termination and padding. To resolve this, the paper introduces Rainbow Padding, a simple yet effective method that reserves a single <eos> for termination and uses a cyclic sequence of <pad_k> tokens for padding. Experiments on models like LLaDA and Dream show that Rainbow Padding effectively eliminates this failure mode, substantially improving performance on reasoning and code generation tasks and restoring length robustness with minimal fine-tuning overhead.

**Strengths:**

- The paper defines and analyze a critical failure mode (<eos> overflow) in dLLMs.
-  Rainbow Padding is a simple but effective solution. Its minimal computational overhead (demonstrated by the efficient LoRA adaptation) makes it a very practical method.
- The experiments are thorough and convincing. The performance improvements on length-sensitive tasks like MATH (e.g., from 0.9% to 34.3% on LLaDA) provide evidence of the method's efficacy. The validation across multiple models, tasks, and decoding strategies demonstrates its robustness.

**Weaknesses:**

- The evaluations on the length-sensitive MATH and GSM8K benchmarks were performed on randomly sampled subsets (>100 problems each) rather than the full test sets. I would like to see deterministic results on the full benchmarks, which may be helpful for future comparisons.
- Alternative Padding Schemes: The paper argues that a single <pad> token would reintroduce the problem of probability concentration. Did you experiment with any non-cyclic, deterministic schemes or a simple random sampling of padding tokens from a small, dedicated set? A empirical comparison might further strengthen the case for the cyclic approach.
- Generality Beyond Instruction-Tuning: Does the <eos> overflow issue also appear in pre-trained dLLMs as well? Is Rainbow Padding suitable for all stages of dLLM training, or specific for the instruction-tuning stage?
- Choice of K: The ablation on K (Table 4) is very insightful, showing a plateau around K=7. Is there a theoretical or heuristic basis for choosing an optimal K? For instance, could it be related to the vocabulary size, model capacity? Or is this a hyperparameter that one must tune empirically for each model?

**Questions:**

Please refer to weaknesses.

---

> ### Author Response · Authors · 2025-11-19
> **Response to Reviewer f5z9 (1/2): W1, W2**
>
> **Response to W1: Evaluation uses randomly sampled subsets (MATH/GSM8K).**
>
> Thank you for this important question. We've now completed evaluations on the full GSM8K dataset (1,319 examples) and a substantially larger subset of the MATH dataset (1,050 examples, obtained by random sampling 150 examples across seven categories from the full 5,000-example dataset).
>
> Due to the significant computational requirements of generating long sequences (1024 tokens) with bidirectional (non-causal) attention in dLLMs, we used a smaller random subset during the submission phase. However, our extended evaluations on LLaDA confirm the robustness of our original findings, supporting the effectiveness of Rainbow Padding.
>
> | **Dataset** |        **Metric**        |    **Paper Reported**    |    **Extended Evaluation**    |
> |-------------|---------------------------|-------------------------------|--------------------------------|
> | MATH    | Acc.                  | 34.3                        | 30.4                         |
> |             | Response Length           | 285.7                        | 291.7                         |
> | GSM8K   | Acc.                  | 79.6                        | 75.5                         |
> |             | Response Length           | 115.7                        | 125.2                         |
>
>
> These results show that the statistics reported in the manuscript are stable and unbiased even when evaluated on significantly larger samples. We will update the manuscript to include full coverage in the final version.
>
> **Response to W2: Compare to Alternative Padding Schemes.**
>
> 1. Uniform Random Padding fails.
>
> Thank you for the valuable suggestion. As the reviewer noted, Uniform Random Padding—sampling padding tokens from a uniform distribution—can also distribute probability mass evenly across tokens. However, as discussed in Section 4 (lines 304–311), this approach is difficult to train and performs substantially worse than our cyclic pattern.
>
> The key issue is that Uniform Random Padding does not provide a single ground-truth target for each position; even an optimally trained model cannot reliably predict a random sample.
> In contrast, Rainbow Padding uses a deterministic pattern that gives a clear training target, making it easy to learn even with lightweight adaptation such as LoRA.
>
> Due to this instability, the random padding scheme performs poorly in practice. After training LLaDA-Base for three epochs (our default setting) with LoRA, its performance is summarized below:
>
> | Method                 | MATH (Acc. / Response Length) | GSM8K (Acc. / Response Length) |
> |------------------------|---------------------------|-----------------------------|
> | Rainbow Padding        | 34.3 / 285.7              | 79.6 / 115.7                |
> | Uniform Random Padding | 30.4 / 1023.99      | 45.5  / 1023.75                  |
>
> In both tasks, Uniform Random Padding underperforms Rainbow Padding and produces excessively long responses (nearly the full 1024-token budget). These inflated outputs occur because the model generates verbose, repetitive attempts to solve the same problem, reflecting its inability to learn stable termination behavior from random padding signals.
>
> 2. Advantages of cyclic deterministic pattern.
>
> The core idea of the cyclic pattern is to distribute probability mass evenly across padding tokens ($1/K$), preventing any single token from assigning high confidence that could trigger overflow (Figure 9 (a)). This property also makes it broadly compatible with various adaptive decoding strategies (e.g., margin-, entropy-based), as discussed in Section 6 (lines 405–422).
>
> Taken together, this empirical comparison highlights the practical advantages of the cyclic design. Although other deterministic patterns may be possible, our results indicate that the cyclic approach offers a strong balance of learnability, stability, and compatibility with adaptive decoding, making it an effective choice in practice.

---

> > ### Author Response · Authors · 2025-11-19
> > **Response to Reviewer f5z9 (2/2): W3, W4**
> >
> > **Response to W3: Generality of \<eos> overflow beyond Instruction-Tuning**
> >
> > We would like to clarify that the \<eos> overflow issue is specific to the instruction-tuning stage and does not appear in base models. During pre-training, dLLMs—like standard LLMs—are trained on long, continuous text without learning explicit stopping signals. As a result, base models simply generate content tokens up to the maximum allocated length. \<eos> overflow emerges only during instruction-tuning, where the model must learn variable-length generation and acquire a notion of when to stop. In current dLLM practice, \<eos> is also used as a padding token, and this dual role introduces a positional bias that leads to \<eos> overflow.
> >
> > **Response to W4: Guideline on selecting $K$ in Rainbow Padding**
> >
> >
> > We thank the reviewer for the valuable question. Our goal is not to identify a single “optimal’’ $K$. Instead, we would like to clarify that Rainbow Padding works reliably as long as each padding token’s expected probability ($\sim 1/K$) is below the model’s confidence on meaningful content tokens. Once this holds, larger $K$ does not materially change behavior, which explains the saturation at $K=7$.
> >
> > To demonstrate this more concretely, we conducted an additional analysis. We measure the lowest confidence assigned to unmasked content tokens during the first 20 generation steps of LLaDA-Instruct, while making \<eos> to be unselectable. Across tasks, content-token confidence stays well above 0.20:
> >
> > | **Dataset** | **1st Percentile** | **10th Percentile** | **Mean** | **90th Percentile** |
> > |-------------|--------------------|----------------------|----------|----------------------|
> > | MATH    | 0.305              | 0.362                | 0.486    | 0.637                |
> > | GSM8K   | 0.257              | 0.326                | 0.472    | 0.632                |
> > | HumanEval  | 0.225            | 0.303                | 0.422    | 0.554                |
> >
> > For Rainbow Padding to prevent early termination, we require $1/K$ to be below this range.
> >
> > Thus:\
> > $K=7  →  1/K \approx 0.14$ (safe),\
> > $K=20  →  1/K \approx 0.05$ (safe),\
> > $K=3 → 1/K \approx 0.33$ (unsafe, overlaps with content-token confidence).
> >
> > This explains why $K=3$ frequently fails but $K≥7$ is consistently effective, and this saturation behavior observed on LLaDA is expected to generalize to other dLLMs as well. We have added this discussion and the new experimental results to the revised manuscript in (Figure 12, lines 476–485 on page 9).

---

### Official Review · Reviewer_pwWJ · 2025-10-31

**Soundness:** 4
**Presentation:** 3
**Contribution:** 3
**Rating:** 8
**Confidence:** 3

**Summary:**

The paper describes a method for improving instruction-tuned dLLMs. It should be of interest to anyone training such models.

- Identifies and formalizes “eos overflow”: increasing the maximum generation budget paradoxically shortens outputs for IT’d dLLMs.
- Traces the cause to dual use of eos as both stop symbol and padding during IT, which teaches strong tail-position priors for <eos>.
- Shows how any-order decoding heuristics then select high-confidence tail positions early, predicting <eos> and triggering a backward termination cascade.
- Proposes Rainbow Padding: keep a single eos for true termination and fill the padded tail with a deterministic cycle of K distinct pad tokens.
- Demonstrates content-first decoding after the fix, robust gains across tasks/models/decoding rules, and fast convergence of the pad-loss.
- Shows LoRA fine-tuning can retrofit existing IT models to eliminate length collapse.

**Strengths:**

- simple and effective: change in padding semantics + brief fine-tune yields large accuracy and length-robustness gains.
- decoder-agnostic robustness: works with confidence/margin/entropy decoding and without block scheduling.
- compelling mechanistic evidence: visualization of pad confidences, eos tail priors, and unmasking order link cause to effect.

**Weaknesses:**

- some results use random subsets (e.g., MATH/GSM8K) and only two dLLM families; full-set, multi-seed reporting would strengthen claims.
- limited head-to-head vs. stronger decoding-time fixes (e.g., calibrated eos priors, length-control objectives) or matched-budget AR models.

**Questions:**

- are pads new vocab items or reserved rare tokens? How is contamination avoided in prompts/instructions?
- does K=7 remain optimal at longer budgets (2k–4k tokens) and other corpora?
- how does Rainbow compare to (i) eos suppression with calibrated thresholds, (ii) separate <pad> with loss masking only, (iii) weak length-control priors?

---

> ### Author Response · Authors · 2025-11-19
> **Response to pwWJ (1/2): W1, W2, Q3**
>
> **Response to W1: Results rely on random subsets (e.g., MATH/GSM8K) and only two dLLM families; full-set, multi-seed reporting would strengthen the claims.**
>
> Thank you for this important question. We've now completed evaluations on the full GSM8K dataset (1,319 examples) and a substantially larger subset of the MATH dataset (1,050 examples, obtained by random sampling 150 examples across seven categories from the full 5,000-example dataset).
>
> Due to the significant computational requirements of generating long sequences (1024 tokens) with bidirectional (non-causal) attention in dLLMs, we used a smaller random subset during the submission phase. However, our extended evaluations on LLaDA confirm the robustness of our original findings, supporting the effectiveness of Rainbow Padding.
>
> | **Dataset** |        **Metric**        |    **Paper Reported**    |    **Extended Evaluation**    |
> |-------------|---------------------------|-------------------------------|--------------------------------|
> | MATH    | Acc.                  | 34.3                        | 30.4                         |
> |             | Response Length           | 285.7                        | 291.7                         |
> | GSM8K   | Acc.                  | 79.6                        | 75.5                         |
> |             | Response Length           | 115.7                        | 125.2                         |
>
>
> These results show that the statistics reported in the manuscript are stable and unbiased even when evaluated on significantly larger samples. We will update the manuscript to include full coverage in the final version.
>
> Regarding model coverage, we selected LLaDA and Dream as they represent primary open-source dLLM families currently available. While additional dLLMs exist, many are closed-source (e.g., Mercury), task-specific (e.g., DiffuCoder-7B), or direct variants of LLaDA (e.g., LLaDA 1.5, MMaDA). Since Rainbow Padding is architecture-agnostic, we expect it to apply naturally to other dLLM architectures as well.
>
> **Response to W2 & Q3: Limited comparison against heuristic decoding-time fixes.**
>
> Thank you for suggesting comparison on alternative approaches. Rainbow Padding is conceptually distinct from heuristic fixes that manually suppress \<eos> probability during inference. Following the reviewer’s suggestion, we evaluated two inference-time strategies:
>
> - Method-1 (M1): Time-scheduled \<eos> activation:\
> We suppress the \<eos> logit for the first 30% of decoding steps by multiplying it with a time-dependent scaling factor that increases linearly from 0 to 1, thereby preventing early \<eos> activation, and restore the original \<eos> logit thereafter.
>
> - Method-2 (M2): Consensus-based \<eos> activation:\
> We activate \<eos> only when over 90% of masked positions predict it as the top-1 token. Otherwise, its logit is set to −inf to prevent premature ending.
>
> We report results on LLaDA-Instruct with max_length = 1024:
>
> | **Method**        | **MATH** (Acc. /Response Length) | **GSM8K** (Acc. / Response Length) |
> |-------------------|------------------------------|-------------------------------|
> | M1                | 20.0 / 627.9                    | 42.4 / 595.6                     |
> | M2                | 20.9  / 990.3                    | 50.0 / 890.3                     |
> | Rainbow Padding   | 34.3 / 285.7              | 79.6 /115.7               |
>
>
>
> Both M1 and M2 underperform Rainbow Padding on downstream tasks and frequently generate excessively long outputs, such as repeated solution attempts or meaningless character sequences, thereby undermining the utility of instruction-tuned models.
>
> In summary, decoding-time heuristics such as M1 and M2 have fundamental limitations: they rely on sensitive hyperparameters (e.g., 30% thresholds or 90% consensus) that require careful tuning for each task or even each prompt type, often trigger at inappropriate times, and distort the model’s learned probability distribution. In contrast, Rainbow Padding requires only a one-time choice of a sufficiently large $K$, providing a task-agnostic setting that works reliably across diverse datasets by allowing the model to internalize proper termination behavior during training.

---

> > ### Author Response · Authors · 2025-11-19
> > **Response to pwWJ (2/2): Q1, Q2**
> >
> > **Response to Q1: Clarification on padding token types to avoid prompt contamination.**
> >
> > We thank the reviewer for highlighting this valuable point. For LLaDA, we rely on explicit ‘reserved’ tokens already present in the vocabulary; the LLaDA authors specifically note that these spare tokens are intended for introducing additional functionality when needed.
> >
> > For Dream, we use extremely rare, non-interpretable character sequences that effectively never appear in English benchmarks (e.g., strings like "ĠĠĠĠĠĠĠĠĠĠĠĠĠĠĠĠĠĠĠĠĠĠĠĠ..." with >60 repetitions) as noted in Appendix C.1.
> >
> > These choices allow us to avoid modifying the model architecture (e.g., resizing embeddings or output heads), thereby minimizing any disruption to the original model and ensuring that the padding tokens do not contaminate prompts or instructions.
> >
> > **Response to Q2: Whether $K=7$ remains optimal for longer budgets and across other corpora.**
> >
> > 1. Robustness of Rainbow Padding to longer budget
> >
> > Thank you for the insightful question. As requested, we evaluated the robustness of Rainbow Padding under a longer context budget (max_length = 2048) using LLaDA. The results are reported below.
> >
> > | **Dataset** | **Metric**         | **max_length = 1024 (our default)** | **max_length = 2048** |
> > |-------------|---------------------|-------------------------|-------------------------|
> > | MATH        | Acc.            | 34.3                     | 33.3                     |
> > | MATH        | Response Length     | 285.7                    | 385.9                     |
> > | GSM8K       | Acc.            | 79.6                     | 75.8                    |
> > | GSM8K       | Response Length     | 115.7                     | 118.1                    |
> >
> > We observe that LLaDA with Rainbow Padding remains stable even when the allocated length budget is doubled to 2048 tokens, indicating that the model learned proper termination behavior.
> >
> > We would like to also clarify that our default setting $K=7$ wasn’t intended to identify a single “optimal’’ value, but to show that Rainbow Padding works effectively once the expected probability of each padding token ($\sim 1/K$) falls below the model’s confidence on content tokens. We have included additional analysis in the revised manuscript (lines 476-485, Figure 12), where we show any modest values (such as 7, 20) are reliable.
> >
> > 2. Performance of Rainbow Padding on other dataset
> >
> > We further evaluate Rainbow Padding on an additional task (MBPP), using a randomly selected subset of 100 samples with max_length=1024.
> >
> > | **Dataset** | **Metric**           | **Rainbow Padding ($K=7$)** | **LLaDA-Instruct** |
> > |---------|------------------|---------------------------|----------------|
> > | MBPP    | Acc.         | 46.0                       | 6.0            |
> > | MBPP    | Response Length  | 54.73                       | 8.11            |
> >
> > We observe that the model trained with Rainbow Padding maintains strong performance on this additional benchmark, whereas the original model degrades substantially due to \<eos>-overflow. This result suggests that Rainbow Padding is expected to naturally generalize well to other downstream tasks.

---

### Official Review · Reviewer_29oR · 2025-10-31

**Soundness:** 2
**Presentation:** 2
**Contribution:** 2
**Rating:** 4
**Confidence:** 2

**Summary:**

This paper identifies and analyzes a major failure mode in dLLMs known as eos overflow, where allocating longer generation lengths paradoxically causes models to terminate earlier or output streams of eos tokens. The authors trace the issue to the dual use of the eos token as both a true sequence terminator and padding during instruction-tuning, which biases the model toward premature endings. To address this, they propose Rainbow Padding, a simple yet effective fix that replaces repeated eos paddings with a cyclic set of distinct padding tokens, thereby distributing probability mass and decoupling padding from termination. Experiments across reasoning and code generation benchmarks show that Rainbow Padding eliminates early termination, improves accuracy and length robustness, and can be applied efficiently via lightweight fine-tuning.

**Strengths:**

* **Originality**: The paper is original in systematically identifying, naming, and diagnosing the critical eos overflow failure mode in diffusion LLMs.
* **Quality**: The claims are supported by empirical evidence and thorough ablations, reliably demonstrating substantial performance gains across multiple models and benchmarks.
* **Clarity**: The paper is well-written, logically structured, and explains complex ideas using descriptive names and visual aids.
* **Significance**: This work addresses a reliability issue that previously hampered the practical utility of instruction-tuned dLLMs.

**Weaknesses:**

- Evaluation inconsistency: The paper reports that “MATH and GSM8K use randomly sampled subsets (>100 problems each).” Such random sampling can lead to unstable or non-reproducible results. It is recommended to evaluate on fixed and standardized test sets (e.g., the MATH-500 subset) to ensure reproducibility and fair comparison across methods.

- Insufficient experimental coverage: The experiments focus solely on fine-tuning from Base models. It remains unclear whether the proposed method generalizes to instruction-tuned models (e.g., LLaDA-Instruct). Additional experiments on such models would strengthen the claim of universality and demonstrate the method’s applicability to real instruction-following settings.

**Questions:**

The current analysis presents $K=7$ as a sufficient number of padding tokens, but the paper does not fully detail a principled or efficient method for determining this optimal $K$ valuefor dLLMs. We ask the authors to provide a clearer discussion or heuristic on how the smallest effective value of $K$ can be efficiently determined. Furthermore, please clarify whether the saturation point observed for LLaDA is expected to be a universal property that can generalize to other dLLMs.

---

> ### Author Response · Authors · 2025-11-19
> **Response to Reviewer 29oR**
>
> **Response to W1: Recommend to evaluate on standardized test sets (MATH/GSM8K).**
>
> Thank you for this important question. We've now completed evaluations on the full GSM8K dataset (1,319 examples) and a substantially larger subset of the MATH dataset (1,050 examples, obtained by random sampling 150 examples across seven categories from the full 5,000-example dataset).
>
> Due to the significant computational requirements of generating long sequences (1024 tokens) with bidirectional (non-causal) attention in dLLMs, we used a smaller random subset during the submission phase. However, our extended evaluations on LLaDA confirm the robustness of our original findings, supporting the effectiveness of Rainbow Padding.
>
> | **Dataset** |        **Metric**        |    **Paper Reported**    |    **Extended Evaluation**    |
> |-------------|---------------------------|-------------------------------|--------------------------------|
> | MATH    | Acc.                  | 34.3                        | 30.4                         |
> |             | Response Length           | 285.7                        | 291.7                         |
> | GSM8K   | Acc.                  | 79.6                        | 75.5                         |
> |             | Response Length           | 115.7                        | 125.2                         |
>
>
> These results show that the statistics reported in the manuscript are stable and unbiased even when evaluated on significantly larger samples. We will update the manuscript to include full coverage in the final version.
>
>
> **Response to W2: Unclear whether Rainbow padding generalized to instruction-tuned models**
>
>
> We thank the reviewer for raising this point and clarify that our experiments already evaluate the utility of Rainbow Padding both on base models and as a post-hoc adaptation applied to instruction-tuned dLLMs.
>
> 1. Instruction-tuning from base:\
> We apply Rainbow Padding directly into the instruction-tuning stage of base models (LLaDA-Base, Dream-Base). This setup shows that Rainbow Padding prevents early termination when constructing instruction-tuned models.
>
> 2. Adapting existing instruction-tuned models:\
> We apply a lightweight 1-epoch LoRA fine-tune to already instruction-tuned models (LLaDA-Instruct, Dream-Instruct). These experiments, reported in Section 6 (lines 424–452), demonstrate that Rainbow Padding can also be applied post hoc to repair early termination in already instruction-tuned models.
>
>
> Overall experimental results for both settings, along with qualitative examples in Appendix D.4 (Figures 22–25), show that Rainbow Padding serves as a simple modification to the instruction-tuning procedure that both prevents early termination and, when necessary, corrects it in already instruction-tuned dLLMs.
>
>
> **Response to Q1: Provide a clearer discussion or heuristic on determining $K$**
>
> We thank the reviewer for the valuable question. Our goal is not to identify a single “optimal’’ $K$. Instead, we would like to clarify that Rainbow Padding works reliably as long as each padding token’s expected probability ($\sim 1/K$) is below the model’s confidence on meaningful content tokens. Once this holds, larger $K$ does not materially change behavior, which explains the saturation at $K=7$.
>
> To demonstrate this more concretely, we conducted an additional analysis. We measure the lowest confidence assigned to unmasked content tokens during the first 20 generation steps of LLaDA-Instruct, while making \<eos> to be unselectable. Across tasks, content-token confidence stays well above 0.20:
>
> | **Dataset** | **1st Percentile** | **10th Percentile** | **Mean** | **90th Percentile** |
> |-------------|--------------------|----------------------|----------|----------------------|
> | MATH    | 0.305              | 0.362                | 0.486    | 0.637                |
> | GSM8K   | 0.257              | 0.326                | 0.472    | 0.632                |
> | HumanEval  | 0.225            | 0.303                | 0.422    | 0.554                |
>
> For Rainbow Padding to prevent early termination, we require $1/K$ to be below this range.
>
> Thus:\
> $K=7  →  1/K \approx 0.14$ (safe),\
> $K=20  →  1/K \approx 0.05$ (safe),\
> $K=3 → 1/K \approx 0.33$ (unsafe, overlaps with content-token confidence).
>
> This explains why $K=3$ frequently fails but $K≥7$ is consistently effective, and this saturation behavior observed on LLaDA is expected to generalize to other dLLMs as well. We have added this discussion and the new experimental results to the revised manuscript in (Figure 12, lines 476–485 on page 9).

---

> > ### Comment · Reviewer_29oR · 2025-11-22
> >
> > Thanks for the detailed responses.
> >
> > W2 and Q1 are well addressed; the additional experiments and explanations are convincing, and I have raised my score.
> >
> > For W1, I wonder why the evaluation was not done on the official MATH-500 subset. Reporting results on that standardized set would improve reproducibility and comparability.

---

> > > ### Author Response · Authors · 2025-11-25
> > > **Response to Reviewer 29oR**
> > >
> > > We appreciate the reviewer’s suggestion. Our primary goal was to remain consistent with the evaluation protocols of LLaDA and Dream, both of which report results on the MATH dataset rather than MATH500. Since MATH500 is a stratified random subset of size 500 from MATH, we expect the performance trends under Rainbow Padding to be essentially similar. Our additional checks on smaller random subsets (105 and 1050 samples) confirm that Rainbow Padding behaves consistently across subset sizes. For reproducibility and comparability, we will include results on the official MATH500 subset in the final version.

---

### Author Response · Authors · 2025-11-19
**General Response by Authors**

We sincerely thank all reviewers for their thoughtful and positive feedback. Reviewers highlighted the originality of identifying and diagnosing the \<eos>-overflow failure mode (29oR, pwWJ, f5z9, rf99); the simplicity, effectiveness, and practicality of Rainbow Padding (29oR, pwWJ, f5z9, rf99); the minimal computational overhead and efficient LoRA adaptation (pwWJ, f5z9); the robustness across diverse decoding strategies (pwWJ); the thorough and comprehensive experiments (29oR, pwWJ, f5z9, rf99); the substantial performance gains (29oR, f5z9); the mechanistic insights supporting our diagnosis (pwWJ); and the clarity of the presentation (29oR). We are grateful for these encouraging assessments.

Regarding the concerns, we address each point below in detail. We have also uploaded a revised manuscript with all corresponding changes highlighted in blue.

The revision includes:
- Correction of a minor typo (line 469, 471)
- Additional analysis on the number of padding tokens ($K$) (lines 476–485, Figure 12).

All line and section references in our responses correspond to this revised manuscript. We sincerely appreciate the opportunity to further improve our work.

---

### Author Response · Authors · 2025-12-01
**Comment for AC by Authors**

We sincerely appreciate the AC's exceptional effort in supporting the ICLR community and managing this unusual situation caused by the OpenReview incident. Below we provide a brief summary of the reviewers’ comments and our responses for the AC’s convenience.

Reviewers raised two main common questions:

**1. Extended evaluation on full / larger datasets (MATH, GSM8K).**

Our initial submission evaluated Rainbow Padding on random subsets of MATH and GSM8K due to computational constraints. Reviewers questioned whether our findings would generalize to full dataset evaluation.

During the discussion phase, we provided results on the full GSM8K dataset (1,319 examples) and a substantially larger MATH subset (1,050 examples). The accuracy and response length statistics closely matched our originally reported results, confirming that our findings are robust and reproducible at scale.

**2. Rationale for K=7 default and behavior with different numbers of distinct padding tokens.**

Reviewers requested a principled explanation for choosing the number of distinct padding tokens (K) in Rainbow Padding.
We emphasize that our default setting K=7 is not a sensitive hyperparameter—any moderately sized K works effectively with minimal overhead.

To elaborate, during the discussion phase we conducted an additional analysis in Section 6 (Figure 12, Table 4), showing that Rainbow Padding remains effective whenever the expected padding-token probability (≈1/K) stays below the content-token confidence during early decoding.

Our empirical analysis shows that content-token confidence typically exceeds 0.20 across all downstream tasks. This threshold informs our design:

- K=7 yields 1/7≈0.14, consistently staying below the content-token confidence threshold.

- K=3 yields 1/3≈0.33, which overlaps with content-token confidence and frequently fails

These results explain why performance saturates at K≥7 in typical tasks, with larger K values yielding only marginal gains, and why smaller K values (e.g., K=3) fail. In summary, any sufficiently large K works reliably well in practice.

During the discussion phase, reviewer 29oR responded positively, finding our explanations convincing and noting that we effectively addressed the above concerns.

Beyond these two common questions, we have addressed all remaining reviewer concerns through manuscript revisions and our discussion-phase responses. We sincerely appreciate this valuable opportunity to strengthen our work.

---

### Meta-Review · Area_Chair_CJqe · 2026-01-06

**Summary:**

The key decision factors were (i) whether the paper convincingly identifies and diagnoses a core failure mode in instruction-tuned diffusion LLMs (ii) whether the proposed Rainbow Padding mechanism provides a simple, effective, and generalizable fix with minimal overhead, and (iii) whether the empirical evaluation is sufficiently robust across tasks, models, and decoding strategies. Reviewers broadly agreed that the paper clearly isolates an important and previously under-analyzed reliability issue that directly affects practical deployment of instruction-tuned dLLMs. The proposed solution is conceptually simple, easy to implement, and empirically effective, with consistent gains in both accuracy and length robustness across reasoning and coding benchmarks. After rebuttal, the evaluation scope and design choices were substantially strengthened, leading to overall consensus in favor of acceptance.

**Reviewer Concerns:**

Addressed concerns:
The rebuttal significantly expanded the empirical evaluation, including results on the full GSM8K dataset and substantially larger MATH subsets, confirming that the originally reported improvements are stable and reproducible at scale. The authors provided a principled explanation for the choice of the number of padding tokens, showing that Rainbow Padding works reliably as long as the per-token padding probability stays below typical content-token confidence, which explains the observed saturation around $K\ge7$. Additional experiments demonstrated robustness to longer generation budgets, applicability to both base and instruction-tuned models (including post-hoc repair via lightweight LoRA fine-tuning), and clear advantages over decoding-time heuristics such as <eos> suppression or consensus-based termination. Comparisons with alternative padding schemes (e.g., uniform random padding) further clarified why the cyclic deterministic design is important for learnability and stability.

Remaining concerns:
Some reviewers noted that the paper still lacks a fully formal theoretical analysis of the training dynamics behind Rainbow Padding, and that baseline coverage could be further broadened. However, these concerns were generally viewed as secondary, given the clarity of the mechanistic explanation, the strength of the empirical evidence, and the practical relevance of the identified failure mode and solution.

**Reviewer Scores:**

29oR: likely 6
pwWJ: likely 8
f5z9: likely 6
rf99: likely 6

---

### Decision · Program_Chairs · 2026-01-26

Accept (Poster)